# Looking for a Treatment for the Early Stage of Alzheimer’s Disease: Preclinical Evidence with Co-Ultramicronized Palmitoylethanolamide and Luteolin

**DOI:** 10.3390/ijms21113802

**Published:** 2020-05-27

**Authors:** Roberta Facchinetti, Marta Valenza, Maria Rosanna Bronzuoli, Giorgia Menegoni, Patrizia Ratano, Luca Steardo, Patrizia Campolongo, Caterina Scuderi

**Affiliations:** 1Department of Physiology and Pharmacology “V. Erspamer”, SAPIENZA University of Rome, P.le A. Moro 5, 00185 Rome, Italy; roberta.facchinetti@uniroma1.it (R.F.); martavalenza@gmail.com (M.V.); mariarosanna.bronzuoli@uniroma1.it (M.R.B.); giorgiamenegoni95@gmail.com (G.M.); patrizia.ratano@uniroma1.it (P.R.); luca.steardo@uniroma1.it (L.S.); patrizia.campolongo@uniroma1.it (P.C.); 2Epitech Group SpA, Saccolongo, 35030 Padova, Italy; 3Università Telematica Giustino Fortunato, Via Raffaele Delcogliano 12, 82100 Benevento, Italy; 4Neurobiology of Behavior Laboratory, Santa Lucia Foundation, 00179 Rome, Italy

**Keywords:** astrocytes, beta amyloid, luteolin, microglia, neuroinflammation, neuroprotection, palmitoylethanolamide, preclinical Alzheimer’s disease, prodromal Alzheimer’s disease, reactive gliosis

## Abstract

Background: At the earliest stage of Alzheimer’s disease (AD), although patients are still asymptomatic, cerebral alterations have already been triggered. In addition to beta amyloid (Aβ) accumulation, both glial alterations and neuroinflammation have been documented at this stage. Starting treatment at this prodromal AD stage could be a valuable therapeutic strategy. AD requires long-term care; therefore, only compounds with a high safety profile can be used, such as the new formulation containing palmitoylethanolamide and luteolin (co-ultra PEALut) already approved for human use. Therefore, we investigated it in an in vivo pharmacological study that focused on the prodromal stage of AD. Methods: We tested the anti-inflammatory and neuroprotective effects of co-ultra PEALut (5 mg/Kg) administered for 14 days in rats that received once, 5 µg Aβ_(1–42)_ into the hippocampus. Results: Glial activation and elevated levels of proinflammatory mediators were observed in Aβ-infused rats. Early administration of co-ultra PEALut prevented the Aβ-induced astrogliosis and microgliosis, the upregulation in gene expression of pro-inflammatory cytokines and enzymes, as well as the reduction of mRNA levels BDNF and GDNF. Our findings also highlight an important neuroprotective effect of co-ultra PEALut treatment, which promoted neuronal survival. Conclusions: Our results reveal the presence of cellular and molecular modifications in the prodromal stage of AD. Moreover, the data presented here demonstrate the ability of co-ultra PEALut to normalize such Aβ-induced alterations, suggesting it as a valuable therapeutic strategy.

## 1. Introduction

Alzheimer’s disease (AD) is a multifactorial neurodegenerative disease of the brain, clinically characterized by progressive memory loss and cognitive decline, leading to dementia and functional disability [1,2]. The onset of AD is insidious since clinical symptoms manifest years after structural alterations of the brain have already been established. At a molecular level, AD lesions consist of plaques of beta amyloid (Aβ) peptides in the extracellular space and neurofibrillary tangles of hyperphosphorylated tau proteins inside neurons. The brain immune system recognizes the abnormal accumulation of proteins as injurious stimuli and reacts to defend the brain [3,4]. Indeed, the neuroinflammatory process begins at the earliest pre-symptomatic phase of AD [5,6,7,8,9]. This process is finely tuned by glial cells, specifically astrocytes and microglia, that quickly respond to all kinds of brain injuries in a complex way, both time- and context-dependent [10]. The response of glial cells to brain perturbance is actually heterogeneous; it can span from reactivity (e.g., microgliosis/astrogliosis) to degeneration (atrophy with subsequent loss of function) [11,12,13,14]. 

Despite the massive scientific efforts, which have been crucial to unveil the molecular mechanisms and the genetics beyond AD, no effective treatments are available for patients. Currently, approved therapies for AD provide only symptom management, resulting in modest or transient benefits to just a subset of patients. Disappointing results of the latest clinical trials have prompted researchers to rethink possible pharmaceutical targets and therapeutic approaches [15]. More and more scientists concur with the notion that the best time for intervention could be the stage at which the disease is not fully overt [16]. In accordance, the 2011 guidelines published by the National Institute on Aging (NIA) prompted scientists to study the earliest stage of dementia, named prodromal AD, when the cellular and molecular alterations leading to the disease have already been triggered, but the affected person is still asymptomatic, therefore, independent and also unaware of his clinical condition [17,18,19]. 

Aβ accumulation has been documented in asymptomatic patients [20,21,22]. Preclinical and clinical investigations have documented foci of activated microglia and astrocytes even before Aβ plaques [23,24,25]. Clinical studies have reported higher expression of markers of astrogliosis and microgliosis, as well as of both peripheral and central neuroinflammation, in young AD cases as compared with old ones, indicating that all these phenomena are prominent at the earliest AD stage, and can decrease with aging [6,14,26,27,28,29]. In line with this evidence, we demonstrated the presence of glial reactivity and neuroinflammation in young 3×Tg-AD mice, whereas old AD transgenic mice exhibited glial atrophy and no signs of neuroinflammation [30]. Therefore, the prodromal stage of AD seems to be characterized by an acute neuroinflammatory burst that wanes, but endures, as the disease progresses. Further studies are needed to clarify whether neuroinflammation could represent an early biomarker for AD. The change in the neuroinflammatory process during the development of AD could be the reason for the failure of the clinical trials testing nonsteroidal anti-inflammatory drugs (NSAIDs), in which patients with fully overt AD were recruited [31]. On the basis of these considerations, dampening glial reactivity and neuroinflammation at the very earliest stage of the disease, and not once AD is fully developed, seems to represent a valuable therapeutic approach [32,33,34,35]. Compounds able to do this, constitute a pool of possible therapeutic drugs to study. However, the prodromal AD phase is difficult to diagnose because there are no specific biomarkers able to reliably and timely detect it, and to manage it, since it involves asymptomatic patients [36]. Falling into preventive care, only compounds with a high safety profile could be of interest. Among them, palmitoylethanolamide (PEA), a naturally occurring amide of ethanolamine and palmitic acid, exerts anti-inflammatory and neuroprotective properties in several preclinical models of AD [37,38,39,40,41,42,43]. In parallel, the flavonoid luteolin (3,4,5,7-tetrahydroxyflavone, Lut), found in different edible plants, showed antioxidant, anti-inflammatory, as well as memory-improving properties [44,45,46]. 

Recent studies have reported that a novel formulation containing both PEA and Lut, ultramicronized together at fixed doses (co-ultra PEALut, 10:1 by mass), exhibited good bioavailability [47,48] and was more efficacious than the two compounds alone [49,50,51]. Co-ultra PEALut has already been licensed for human use (Glialia®) as a dietary food for special medical purposes, intended for the dietary management of individuals with central neuroinflammation and associated oxidative stress. It has demonstrated good safety and tolerability, thus, representing a valid tool with translational relevance for a long-lasting treatment as AD requires. Both the single compounds (i.e., micronized-PEA and luteolin) and the co-ultramicronized composite have optimal safety profiles, as evidenced by the lack of toxicity and genotoxic potential [52,53] and few, if ever, adverse effects so far reported in clinical studies [54,55,56,57].

Co-ultra PEALut exerted neuroprotective effects in models of neurological conditions that share some features with AD, such as cerebral ischemia, traumatic brain injuries, and vascular dementia [51,54,58,59]; it also reduced markers of inflammation in an animal model of Parkinson’s disease [60]. Regarding AD, promising results were obtained in vitro showing that co-ultra PEALut treatment of both human neuroblastoma cells and rat organotypic cultures, challenged with Aβ, exerted anti-inflammatory and neuroprotective effects [61]. However, no data are available yet on the effects of co-ultra PEALut administration in an in vivo model of AD. With the aim to contribute by filling this gap and considering that the prodromal stage of AD represents a temporal window in which it could be possible to reduce both risk and incidence of the disease [62,63,64,65,66,67], we designed an in vivo pharmacological study focused on the prodromal stage of AD. We tested the effects of intraperitoneal (i.p.) co-ultra PEALut (5 mg/Kg or vehicle) administration for 14 consecutive days in a validated rodent model of Aβ-induced neurotoxicity [37,68]. Gene and protein expression analyses of markers for astrocyte and microglia activation, as well as for parameters of neuroinflammation, were investigated. Since reduced release of neurotrophic factors has been documented in both pre-symptomatic and full-blown AD patients [69,70], we carried out experiments on expression of both glial-derived and brain-derived neurotrophic factors (GDNF and BDNF). 

All results presented in this study provide the first in vivo evidence for the efficacy of co-ultra PEALut treatment, started as early as Aβ accumulates, in counteracting the molecular alterations induced by Aβ. Our data support a rapid translation into the clinic, since co-ultra PEALut is already approved for human use.

## 2. Results 

### 2.1. Co-Ultra PEALut Prevented the Protracted Activation of Hippocampal Astrocytes and Microglia Induced by Aβ_(1–42)_ Challenge

Aβ accumulation has been documented in asymptomatic patients [21,22], and both histopathological and structural studies have shown that the hippocampus is one of the first brain areas affected [71,72]. Here, we tested the effects of chronic i.p. administration of co-ultra PEALut in modulating astrogliosis and microgliosis induced by a single intrahippocampal Aβ_(1–42)_ peptide infusion. 

Astrocytes provide trophic, metabolic, and synaptic support to neurons, contributing to brain homeostasis and blood–brain barrier structuration [73,74]. Here, we assessed the astrocytic cytoskeletal protein glial fibrillary acidic protein (GFAP) to study the morphological changes occurring on astrocytes upon activation [75,76]. As shown in Figure 1A,B, a significantly stronger GFAP immunofluorescence signal was detected in Aβ_(1–42)_-inoculated rats as compared with the control rats, suggesting that astrocytes were still activated 15 days after a single Aβ peptide infusion. The Aβ_(1–42)_-inoculated rats that received daily systemic co-ultra PEALut showed significantly lower GFAP expression as compared with rats systemically treated with the respective vehicle. 

Microglia are the brain-resident phagocytes and antigen presenting cells [77]. Here, we investigated microglial morphology studying the ionized calcium-binding adapter molecule 1 (Iba1) and the cluster of differentiation (CD) 11b. The first is a marker for microglial cytoplasmic processes, which are the cell sensors for exploring the surroundings [78], whereas CD11b is a marker for both activated local microglia and circulating monocytes, reaching the site of brain injury [74]. 

As shown in Figure 1C,D, Aβ_(1–42)_-inoculated rats showed a higher level of Iba1 protein immunolabeling as compared with their vehicle-infused counterparts. Chronic i.p. administration of co-ultra PEALut completely blocked the raise in Iba1 protein fluorescent signal seen in the Aβ_(1–42)_-infused rats as compared with rats that received systemic vehicle. Moreover, CD11b gene expression was found to be higher in the hippocampi of rats that received a single Aβ_(1–42)_ stereotaxic infusion than in the vehicle-infused rats (Figure 1E). Chronic systemic administration of co-ultra PEALut, but not vehicle, prevented such an increase. Interestingly, co-ultra PEALut effects were observed only in the Aβ_(1–42)_-infused animals.

### 2.2. Co-Ultra PEALut Treatment Prevented the Aβ_(1–42)_-Induced Upregulation of Several Proinflammatory Genes

We examined the possible anti-inflammatory effect of chronically administered co-ultra PEALut on gene expression of several proinflammatory mediators in the hippocampus. In different preclinical models of AD, we have already shown the Aβ_(1–42)_-induced increase in expression level of both inducible nitric oxide synthase (iNOS) and cyclooxygenase (COX)-2, which are two enzymes responsible for NO and prostaglandins production, respectively, as well as of pro-inflammatory cytokines, including interleukin (IL)-1β, IL-6 and tumor necrosis factor (TNF)-α [38,79,80]. Low levels of anti-inflammatory mediators, such as IL-10, has also been documented [81].

Gene expression of both iNOS and COX-2 was significantly higher in rats that received a single intrahippocampal Aβ_(1–42)_ infusion as compared with the vehicle-injected rats, as shown in Figure 2A,D, respectively. Chronic administration of co-ultra PEALut significantly prevented such increase, as compared with Aβ_(1–42)_-inoculated rats that received systemic vehicle.

We found upregulated gene expression of IL-1β, TNF-α, and IL-6 in the hippocampus after cerebral infusion of Aβ_(1–42)_, and not of vehicle. Chronic co-ultra PEALut treatment was able to prevent such upregulation (Figure 2B,C,E). The Aβ_(1–42)_-inoculated rats also showed a significantly lower IL-10 mRNA level than their vehicle counterparts, as shown in Figure 2F. Daily co-ultra PEALut, but not vehicle, administration significantly normalized IL-10 gene expression. Co-ultra PEALut effects on all these parameters were observed only in Aβ_(1–42)_-infused animals.

### 2.3. Co-Ultra PEALut Promoted Survival of Hippocampal Neurons Impaired by Aβ_(1–42)_ Challenge

To study the possible neuroprotective effect of early treatment with co-ultra PEALut, we labeled cells for microtubule associated protein (MAP)-2, a specific neuronal protein of the cytoskeleton, in the CA1 subregion of the hippocampus. As shown in Figure 3A,B, protein immunoreactivity was significantly lower in Aβ_(1–42)_-inoculated rats as compared with the vehicle-infused rats, possibly suggesting neuronal death. Chronic systemic treatment with co-ultra PEALut, but not vehicle, prevented the reduction of MAP-2 fluorescent signal detected in Aβ_(1–42)_-infused rats.

We also tested the hypothesis that systemic co-ultra PEALut, given concurrently with intracerebral Aβ_(1–42)_ challenge and continued consecutively for two weeks, would prevent the well-documented Aβ_(1–42)_-altered release of neurotrophins [82], including GDNF and BDNF. GDNF, which is produced by both astrocytes and neurons, is involved in neuronal survival and plasticity [83], whereas BDNF is a neurotrophin mainly produced by neurons. However, it has been documented that astrocytes after brain injury can produce BDNF [84] promoting neuronal growth and survival, thus participating in the synaptic processes of memory [82]. As shown in Figure 3C,D, the results from gene expression analysis showed significantly lower levels of both GDNF and BDNF mRNA in the hippocampus of Aβ_(1–42)_-inoculated rats than that of the vehicle-infused animals. Chronic treatment with co-ultra PEALut, but not vehicle, significantly prevented the reduced gene expression of both neurotrophins in the Aβ_(1–42)_-infused rats. Once again, co-ultra PEALut effects were observed only in the Aβ(_1–42)_-infused animals.

## 3. Discussion 

To date, none of the drugs approved for AD are effective [2,63], thus novel therapeutic strategies are needed. The reason for failure of many therapeutics against AD could lie in the timing of the start of treatments, as usually AD is already full-blown [85]. However, nowadays, it is known that AD molecular alterations are present decades before the fully overt stage of the disease. Accordingly, the 2011 NIA guidelines recommend scientists to deeply characterize the asymptomatic phase of AD, which starts several years before the onset, suggesting it as the best window for treatment [17]. To note, clinical evidence highlights the presence of glial activation even before Aβ plaques formation, as well as signs of neuroinflammation at the earliest stage of AD, while these phenomena seem to be mitigated as the pathology progresses [6,24,25,26,27,28,29]. However, starting a pharmacological treatment at this early stage opens two key issues for clinicians as follows: First, the need of biomarkers, not available so far, to diagnose the prodromal AD phase and to distinguish normal aging from the pathological one [19]; and second, the need to administrate treatments to patients not experiencing any symptom. Such a pharmacological approach falls into the preventive care field and requires an extremely high level of safety profile. Keeping these points in mind, we tested the effects of a new formulation containing both PEA, a well-known anti-inflammatory and neuroprotective compound, and the antioxidant Lut, ultramicronized together. The safety of this product, even for prolonged use, has already been demonstrated, thus, representing an ideal candidate for AD management starting from the preclinical phase. To test the effects of co-ultra PEALut in the asymptomatic phase, we decided to use a preclinical model of AD extensively characterized in our laboratory [37,68,79,86]. Adult rats, challenged once with 5 μg human fibrillary Aβ_(1–42)_ peptide in the hippocampus, show mild behavioral deficits 21 days after infusion [37]. This behavioral impairment can be partially matched to working-like memory seen in humans [87,88], which has been considered an early marker for AD [89,90]. Therefore, using the same animal model and to be sure to study an asymptomatic stage, we decided to restrict our time point of observations to 15 days. 

The present results revealed the presence of astrogliosis and microgliosis in the rat hippocampi 15 days after Aβ_(1–42)_ challenge, detected as increased immunofluorescent cells labeled with GFAP and Iba1, respectively, in line with both clinical and preclinical reports documenting the presence of glial activation and neuroinflammation in the prodromal phase of AD [27,28,30,91,92,93]. In accordance, we found upregulated gene expression of both iNOS and COX-2 enzymes, as well as of TNF-α, IL-1β, and IL-6, in Aβ_(1–42)_ rats as compared with the control vehicle-inoculated rats. IL-1β is critically involved in AD pathophysiology since its production, stimulated by Aβ, promotes glial activation perpetuating its further release [94,95,96]. The findings presented in this study support our current hypothesis that glial alterations and neuroinflammation could represent early signs in AD. 

A growing body of evidence shows that PEA, an endogenous lipid messenger produced on demand by glial cells, displays analgesic, antidepressant, anti-inflammatory, and neuroprotective effects in several preclinical models of a variety of diseases, including AD [37,97,98,99,100]. Our group and others have also demonstrated a mechanism by which PEA modulates its actions involving the peroxisome proliferator-activated receptor (PPAR)-α [101,102,103,104,105,106,107,108]. To overcome the limits for administration imposed by its lipid structure, PEA has been ultramicronized and tested for oral bioavailability and pain relieving effect, demonstrating both superior absorption and efficacy as compared with naïve formulations [48,109]. Recently, a different formulation has been synthetized where PEA was ultramicronized together with the flavonoid Lut. Superior properties of co-ultra PEALut as compared with the two compounds alone has been suggested [49,51,110]. This is probably related to the inhibition of the PEA crystallization process performed by Lut, ultimately leading to highly stable microparticles [111]. In both human neuronal cells and in rat hippocampal slice cultures challenged with Aβ, the treatment with co-ultra PEALut exerted anti-inflammatory and neuroprotective effects [61]. In accordance with this in vitro evidence, our results show the beneficial effects of co-ultra PEALut treatment in counteracting Aβ_(1–42)_-induced prolonged neuroinflammation and glial activation. The present findings are the first in vivo evidence using co-ultra PEALut in a preclinical model of AD. Taken together with already published data [30], they reinforce our hypothesis that formulations containing PEA could be beneficial in AD, especially at its earliest stages.

In our experimental condition, co-ultra PEALut was able to also exert neuroprotective effects. Indeed, similar to the reduction of BDNF and GDNF documented in prodromal AD patients [69,70], in the currently used animal model we observed decreased gene expression levels of these two neurotrophins in the hippocampus of Aβ_(1–42)_-inoculated rats as compared with the control vehicle-infused animals. This neuroprotective effect exerted by chronic co-ultra PEALut is of particular importance, since BDNF and GDNF are key neurotrophins regulating neuronal branching, synaptic plasticity, and supporting the growth and survival of several neuronal populations [112]. In accordance with the present data, Li et al. observed that PEA treatment was able to revert the decrease in BDNF and GDNF concentrations in the hippocampus through the participation of the PPARα signaling pathway, in a rodent model of stress-induced depression [99]. As a consequence of the poor neurotrophic support, intrahippocampal Aβ_(1–42)_ injection was sufficient to induce neuronal damage in our experimental condition, detected as lower immunofluorescent signal of the neuronal marker MAP-2. Treatment with co-ultra PEALut prevented it, thus promoting neuronal survival, as suggested by the normalized level of MAP-2-positive signal found in Aβ_(1–42)-_infused rats chronically treated with co-ultra PEALut instead of vehicle. In line with an in vitro investigation [113], our results enlarge the body of evidence on the neuroprotective effects of co-ultra PEALut. 

Taken together, our in vivo data demonstrate the anti-inflammatory and neuroprotective effects of co-ultra PEALut administered timely in a preclinical model of AD, at an asymptomatic stage. The mechanism of action underlying the synergy of the association of PEA with Lut remains to be elucidated. Despite this, the translational value of the present study is valuable. Indeed, co-ultra PEALut is already approved for human use as dietary food for special medical purposes (Glialia®). One report has already tested Glialia® (700 mg PEA/70 mg Lut, daily for nine months) in an asymptomatic 67-year-old woman affected by mild cognitive impairment, the clinical stage immediately before full-blown AD, resulting in a significant improvement of her neuropsychological performances [56]. In accordance, our in vivo preclinical results impel towards a rapid translation of them into the clinical practice as an early therapeutic strategy for AD.

## 4. Material and Methods

### 4.1. Animals 

Adult male Sprague-Dawley rats (250–275 g at the time of surgery; Charles River Laboratories, Calco, Italy) were individually housed in a 12 h light/dark cycle (lights ON at 7 a.m., OFF at 7 p.m.) behavioral facility, in controlled conditions (20 ± 1 °C temperature), with ad libitum food and water. 

All procedures involving animal care or treatments were approved by the Animal Care and Use Committee of the Italian Ministry of Health (prot. 19/1/2012) and performed in compliance with the guidelines of the Directive 2010/63/EU of the European Community Council. All efforts were made to minimize animal suffering and to reduce the number of animals used.

### 4.2. Surgical Procedures 

Stereotaxic surgeries and in situ Aβ_(1–42)_ injection were performed as previously published [37,68]. Briefly, rats (*n* = 4–5 for each experimental group) were anesthetized with sodium pentobarbital 50 mg/Kg i.p. and placed in a stereotaxic frame. An injector, connected to a tubing linked to a microsyringe mounted on a microdialysis pump, was lowered to reach the dorsal hippocampus, using the following coordinates, relative to the bregma: AP −3 mm, ML ± 2.2 mm, and DV −2.8 mm [114]. A volume of 2.5 µl containing 5 µg of human fibrillary Aβ_(1–42)_ or artificial cerebrospinal fluid (aCSF) was unilaterally inoculated at a 0.5 µl/min rate. To minimize backflow and facilitate diffusion, the needle was left in place for an additional 5–8 min after injection [68]. 

### 4.3. Drugs and Drug Treatment 

Human Aβ_(1–42)_ was purchased from Tocris Cookson (Bristol, UK). Aβ_(1–42)_ was dissolved in sterile aCSF at the concentration of 2 µg/µL. The solution was incubated at 37 °C for at least 24 h to obtain the peptide in its fibrillary form [37]. Five micrograms of fibrillary Aβ_(1–42)_ were injected unilaterally into the dorsal hippocampus. Vehicle rats were similarly inoculated with an equal volume (2.5 µL) of aCSF, referred in all figures as vehicle (Veh).

A preparation containing co-ultramicronized PEA and Lut (10:1, by mass; kind gift of Epitech Group SpA, Saccolongo, Italy) was dissolved in 10% pluronic F-68 (Sigma-Aldrich, Saint Louis, MO, USA), solubilized in 90% saline, and injected i.p. Starting on the day of surgery, rats were treated i.p. with co-ultra PEALut (5 mg/Kg) once a day for 14 consecutive days. The control rats received i.p. an equivalent volume of vehicle, referred in all figures as Veh. Doses and administration route were chosen according to the literature [115], as well as pilot experiments (data not shown). No adverse reaction was observed at the injection point or anywhere else, as well as no difference in the cumulative 14-day body weight change was detected between vehicle treated and co-ultra PEALut treated rats (Veh group mean ± SEM 84.0 ±12.3; co-ultra PEALut group mean ± SEM 90.4 ± 9.1; *t*(8) = −0.41, *p* = 0.69). No animals died during the experiment. On day 15, 24 h after the last co-ultra PEALut injection, rats were sacrificed, and brains extracted for molecular analyses. 

### 4.4. Immunofluorescence

Immunofluorescence analysis was performed as previously reported [93,116]. Upon rat sacrifice, brains were immediately extracted, flash frozen using 2-methylbutane, and stored at −80 °C [117]. Coronal slices (20 μm thickness) containing the dorsal hippocampal regions were obtained using a cryostat (Thermo Fisher Scientific, Waltham, MA, USA) and immediately mounted on slides. Then, hippocampal slices were fixed through a bath in 4% paraformaldehyde, prepared in 0.1 M phosphate buffer saline (PBS), for 10 min at 4 °C.

Experimental conditions for immunofluorescence staining are summarized in Table 1. Briefly, sections were blocked in a solution of 5% bovine serum albumin (BSA) in PBS containing 0.25% Triton X-100 (PBS/Triton) for 60 min, at room temperature (RT). Slices were incubated overnight at 4 °C, in the same blocking solution in which one of the following primary antibodies was diluted: rabbit anti-GFAP (1:1000, Abcam, Cambridge, UK), rabbit anti-Iba1 (1:1000, Wako, Pure Chemical Industries, Osaka, Japan), and mouse anti-MAP-2 (1:250, Novus Biologicals, Littleton, CO, USA). After rinses (3 × 10 min) with PBS/Triton, sections were incubated in the proper secondary antibody (1:200, fluorescein-affinipure goat anti-rabbit IgG (H+L); 1:300, rhodamine-affinipure goat anti-mouse IgG (H+L), all from Jackson ImmunoResearch, Suffolk, UK) dissolved in fresh BSA-PBS/Triton solution for 2 h RT. Nuclei were stained with Hoechst (Thermo Fisher Scientific) diluted 1:500 in double distilled water. Slides were mounted with Fluoromount aqueous mounting medium (Sigma-Aldrich) and cover slipped. Fluorescent signal was detected by an Eclipse E600 microscope using a Nikon Plan 10X/10.25 and Nikon Plan Fluor 20X/0.5 objectives (Nikon Instruments, Rome, Italy). Pictures were captured using a QImaging camera (Surrey, BC, Canada) with NISelements BR 3.2 64-bit software (Nikon Instruments). Gain and time exposure were kept constant during all image acquisitions to prevent artifacts. Image analysis was performed using Fiji software. Data were expressed as the ratio of the difference between the mean target fluorescence signal and its background (ΔF) to the non-immunoreactive region signal (F_0_).

### 4.5. Real-Time Quantitative Polymerase Chain Reaction (RT-qPCR)

RT-qPCR was performed as previously described [116,118]. Briefly, total mRNA was isolated using TRI-Reagent (Sigma-Aldrich) and quantified by D30 BioPhotometer spectrophotometer (Eppendorf AG, Hamburg, Germany). One microgram of mRNA was reverse transcribed using a first-strand cDNA synthesis kit, in the presence of 0.2 µM oligo(dT) and 0.05 µg/µL random primers (Promega, Promega Corporation, WI, USA). The thermal protocol included a step at 25 °C for 10 min and one at 72 °C for 65 min. Primers were targeted specifically for CD11b, iNOS, IL-1β, IL-6, and IL-10 (Bio-Rad, Hercules, CA, USA) and for COX-2, TNF-α, GDNF, and BDNF (Bio-Fab laboratories, Rome, Italy). All primer sequences and details are listed in Table 2. Primers (500–800 nM) and cDNA (20 ng) were mixed with the iTaq Universal SYBR Green Supermix (Bio-Rad). The thermal protocol used for amplification included an initial step at 95 °C for 3 min and 40 cycles with one step at 95 °C for 10 s and one at 60 °C for 30 s, using the CFX96 Touch thermocycler (Bio-Rad). The melting curve analysis of the amplification products was carried out at the end of the reaction increasing the temperature from 65 to 95 °C with 0.5 °C increasing steps. The amount of each target amplicon was normalized to the mean mRNA of two reference genes, the TATA-box binding protein (TBP), and the hypoxanthine guanine phosphoribosyl transferase (HPRT) (Bio-Fab laboratories). Data are expressed as ΔΔCq, calculated using the CFX Manager software (Bio-Rad). 

### 4.6. Statistical Analysis

Statistical analysis and graphs were performed using GraphPad Prism software version 6.0 (GraphPad Software, San Diego, CA, USA). Data were analyzed by two-way analysis of variance and, upon detection of a main significant effect or interaction, multiple comparisons were carried out by the Bonferroni’s post-hoc test. Differences between mean values were considered statistically significant when *p* < 0.05.

## Figures and Tables

**Figure 1 ijms-21-03802-f001:**
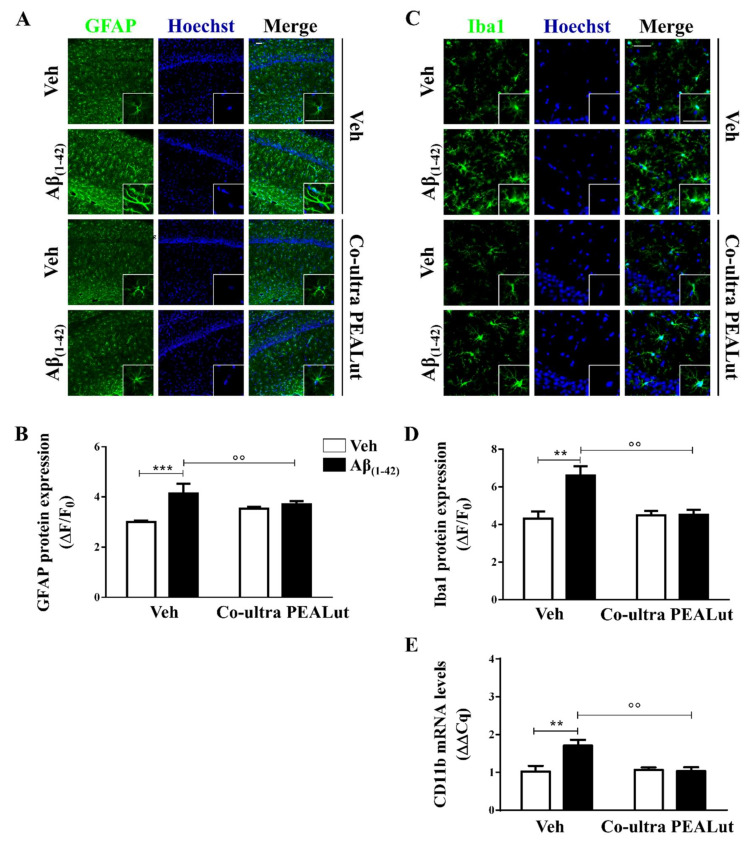
Chronic treatment with 5 mg/Kg co-ultramicronized palmitoylethanolamide and luteolin (co-ultra PEALut) reduced Aβ_(1–42)_-induced astrocyte and microglia activation in the CA1 hippocampal subregion. (**A**) Representative fluorescent photomicrographs of GFAP (green) staining, ipsilateral to the injection site, of Aβ_(1–42)_ (or vehicle)-inoculated rats treated with co-ultra PEALut (5 mg/Kg) or vehicle for 14 day. Nuclei were stained with Hoechst (blue). Scale bar is 50 μm; (**B**) Quantification of GFAP fluorescent signal expressed as the ratio of the difference between the mean fluorescence of the sample and the background (ΔF) to the fluorescence of the non-immunoreactive regions (F_0_); (**C**) Representative fluorescent photomicrographs of Iba1 (green) staining, ipsilateral to the injection site, of Aβ_(1–42)_ (or vehicle)-inoculated rats treated with co-ultra PEALut (5 mg/Kg) or vehicle for 14 day. Nuclei were stained with Hoechst (blue). Scale bar is 30 μm, (**D**) Quantification of Iba1 fluorescent signal expressed as the ratio of the difference between the mean fluorescence of the sample and the background (ΔF) to the fluorescence of the non-immunoreactive regions (F_0_); (**E**) Relative mRNA expression of CD11b, expressed as ΔΔCq, in the hippocampus of both vehicle- and Aβ_(1–42)_-inoculated rats chronically treated with either co-ultra PEALut (5 mg/Kg) or vehicle. Data are presented as mean ± SEM of three independent experiments performed in triplicate. ** *p* < 0.01 and *** *p* < 0.001 versus Veh/Veh; °° *p* < 0.01 versus Aβ/Veh; Bonferroni’s multiple comparisons test.

**Figure 2 ijms-21-03802-f002:**
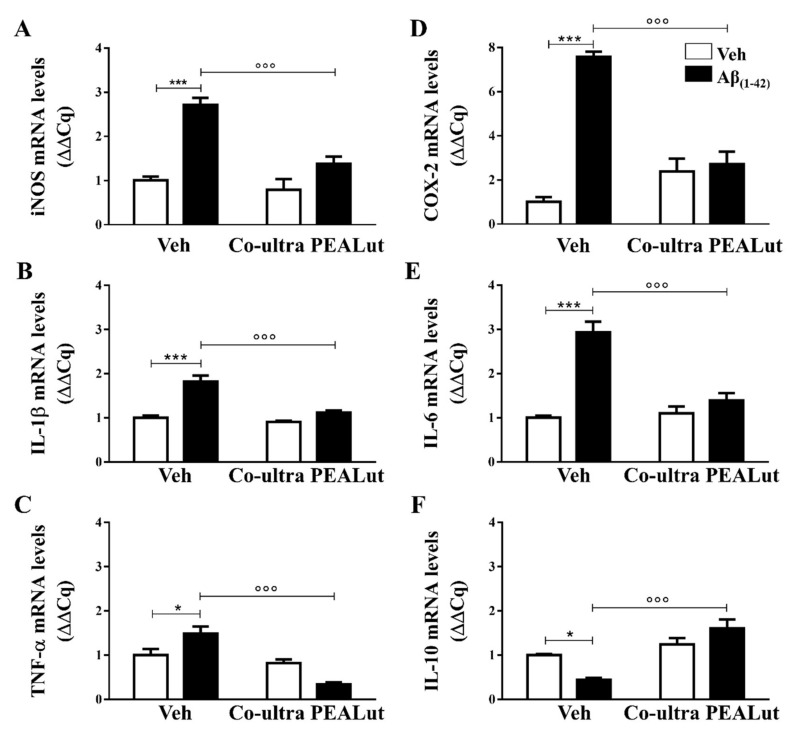
Chronic treatment with co-ultra PEALut blunted gene expression of several markers of neuroinflammation triggered by intrahippocampal Aβ_(1–42)_ injection. Relative mRNA expression of iNOS (**A**), IL-1β (**B**), TNF-α (**C**), COX-2 (**D**), IL-6 (**E**), and IL-10 (**F**) in the hippocampus of rats inoculated with Aβ_(1–42)_, or vehicle, and chronically treated with either co-ultra PEALut (5 mg/Kg/die) or its vehicle. Data are expressed as ΔΔCq and presented as mean ±SEM of four independent experiments performed in triplicate. * *p* < 0.05 and *** *p* < 0.001 versus Veh/Veh; °°° *p* < 0.001 versus Aβ/Veh; Bonferroni’s multiple comparisons test.

**Figure 3 ijms-21-03802-f003:**
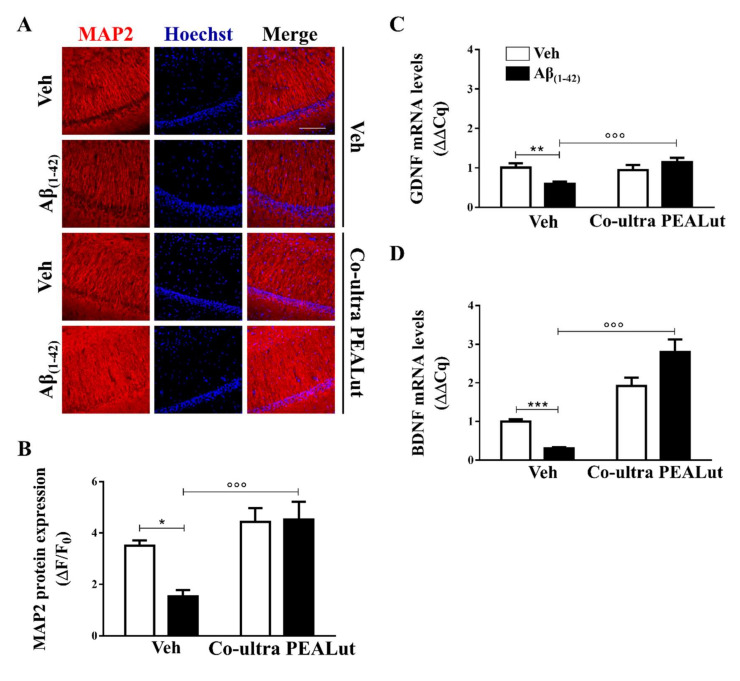
Chronic treatment with co-ultra PEALut promoted neuronal survival impaired by Aβ intracerebral injection and normalized gene expression of neurotrophic factors, lowered by intrahippocampal injection of Aβ_(1–42)_. (**A**) Representative fluorescent photomicrographs of MAP-2 (red) staining in the CA1 hippocampal subregion, ipsilateral to Aβ_(1–42)_ (or vehicle) injection site, obtained from rats chronically treated with either co-ultra PEALut (5 mg/Kg/die) or vehicle. Nuclei were stained with Hoechst (blue). Scale bar is 100 μm; (**B**) Quantification of MAP-2 fluorescent signal expressed as the ratio of the difference between the mean fluorescence of the sample and the background (ΔF) to the fluorescence of the non-immunoreactive regions (F_0_); Gene expression of GDNF (**C**) and BDNF (**D**), expressed as ΔΔCq, in the hippocampus of rats inoculated with Aβ_(1–42)_, or vehicle, chronically treated with either co-ultra PEALut (5 mg/Kg/die) or its vehicle. Data are presented as mean ± SEM of four independent experiments performed in triplicate. * *p* < 0.05, ** *p* < 0.01, and *** *p* < 0.001 versus Veh/Veh; °°° *p* < 0.001 versus Aβ/Veh; Bonferroni’s multiple comparisons test.

**Table 1 ijms-21-03802-t001:** Immunofluorescence experimental conditions.

Primary Antibody	Brand	Dilution	Secondary Antibody	Brand
Rabbit α-GFAP	Abcam	1:200 5% BSA in PBS/0.25% triton X-100	FITC conjugated goat anti-rabbit IgG (H+L) 1:200, 5% BSA in PBS/0.25% triton X-100	Jackson ImmunoResearch
Rabbit α-Iba1	Wako	1:1000 1% BSA in PBS/0.25% triton X-100	FITC conjugated goat anti-rabbit IgG (H+L) 1:200, 0.5% BSA in PBS/0.25% triton X-100	Jackson ImmunoResearch
Mouse α-MAP-2	Novus Biologicals	1:200 5% BSA in PBS/0.25% triton X-100	TRITC conjugated goat anti-mouse IgG (H+L) 1:200, 0.5% BSA in PBS/0.25% triton X-100	Jackson ImmunoResearch

**Table 2 ijms-21-03802-t002:** List of primer sequences, general conditions, and validation parameters used to perform RT-qPCR.

Gene		Primer (5’ → 3’)	Annealing (°C)	Efficiency (%)	R^2^
CD11b	Forward	N/A (Cod. qRnoCID0002800, Bio-Rad)	60	94.0	0.990
	Reverse				
iNOS	Forward	N/A (Cod. qRnoCED0020417, Bio-Rad)	60	98.0	0.999
	Reverse				
COX-2	Forward	GATGACGAGCGACTGTTCCA	60	99.7	0.991
	Reverse	TGGTAACCGCTCAGGTGTTG			
IL-1β	Forward	N/A (Cod. qRnoCID0004680, Bio-Rad)	60	98.0	0.999
	Reverse				
IL-6	Forward	N/A (Cod. qRnoCID0053166, Bio-Rad)	60	94.0	0.998
	Reverse				
TNF-α	Forward	CCACCACGCTCTTCTGTCTA	60	104.7	0.984
	Reverse	CTTGTTGGGACCGATC ACCC			
IL-10	Forward	N/A (Cod. qRnoCID0005930, Bio-Rad)	60	98.0	0.999
	Reverse				
GDNF	Forward	CACCAGATAAACAAGCGGCG	60	99.8	0.989
	Reverse	TCGTAGCCCAAACCCAAGTC			
BDNF	Forward	GGGACTCTGGAGAGCGTGAA	60	103.8	0.996
	Reverse	GTCAGACCTCTCGAACCTGC			
HPRT	Forward	TCCCAGCGTCGTGATTAGTGA	60	98.3	0.992
	Reverse	CCTTCATGACATCTCGAGCAAG			
TBP	Forward	TGGGATTGTACCACAGCTCCA	60	99.7	0.995
	Reverse	CTCATGATGACTGCAGCAAACC			

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
