# Peer review of "Looking for a Treatment for the Early Stage of Alzheimer’s Disease: Preclinical Evidence with Co-Ultramicronized Palmitoylethanolamide and Luteolin"

_ijms, 2020, doi:10.3390/ijms21113802_

Round 1
Reviewer 1 Report
The present article by Facchinetti et al. is entitled “Looking for a treatment for the early stage of AD: preclinical evidence with co-ultramicronized palmitoylethanolamide and luteolin”. This pharmacological research reports first in vivo evaluations of co-ultraPEALut, a nutritional complement composed of ultramicronized N-palmitoylethanolamide and Luteolin, against biological signs of neuroinflammation displayed in early stage of AD. For the team, this study appears of a next step before evaluation in human of co-ultra PEALut in Alzheimer’s Disease.
This article begins with an introduction which state clearly the background (AD, physiopathology, neuroinflammation…) and the rational of the study as well as the expertise of the team in the field from technical and theoretical point of view.
Authors could explain why they choose i.p. administration rather than oral administration (1 sentence).
Results section is well written, sound and concise, maybe too short. Because no study showed tolerance of the i.p administration of co-ultraPEALut in rat, a brief section could be added comparing vehicule versus co-ultraPEALut alone for biological parameters assayed here and for classical in vivo parameters (i.e rat weight during experiences, lethality, observations at the i.p. injection points…). Tolerability is the second key concept of the rationale.
The discussion part is convincing, pursuing the good quality of the paper.
The experimental section is brief and supported by appropiate self-citation. My experience about biological technics is not good enough to review step by step this part.
To conclude, the research presented in this paper is interesting, well build and correctly written. The rational is convincing. Some very minor suitable corrections may improve the text:
L57: The section focused on AD physiopathology mechanism hypothesis should be separate from “Aβ accumulation…”
L83: what are the medical purposes for glialia ?
L95 Author should add “i.p. administration”
L112 role of astrocyte may include BBB structuration
L551, L574 : DOI is missing. Please check all the bibliographic section
Figures :
The legend of “veh/Aβ” using white or black rectangles is present just in Fig 1C and in Fig 3C, is missing in Fig 2. It should be easier to read aided with clear legend common for all the histograms.
Fig 1E : Y-axe should be normalized at 4 ΔΔCq
Author Response
We thank the Reviewer for the comments and appreciate her/his contribution to the improvement of the manuscript. We have modified the manuscript according to her/his criticisms. Please see the changes made in the main text as tracked change. We believe our manuscript has been greatly improved and we acknowledge the contribution of Reviewer#1.
Authors could explain why they choose i.p. administration rather than oral administration (1 sentence).
We have chosen the i.p. administration route for few reasons. Our previous pilot experiments to assess the effective dose of co-ultra PEALut in our condition were carried out using i.p. administration since we have extensive experience with delivering palmitoylethanolamide using that route. The i.p. route allowed us to deliver a precise dosage into the rat with minimal stress; however, future studies will benefit from using oral administration since it is the golden standard for translational purposes. We have clarified this issue in the section “Drugs and drug treatments".
Results section is well written, sound and concise, maybe too short. Because no study showed tolerance of the i.p administration of co-ultraPEALut in rat, a brief section could be added comparing vehicule versus co-ultraPEALut alone for biological parameters assayed here and for classical in vivo parameters (i.e rat weight during experiences, lethality, observations at the i.p. injection points…). Tolerability is the second key concept of the rationale.
According to the Reviewer’s suggestions, we have added a sentence in the section “Drugs and drug treatments" to report that no animal died during the experiment or following co-ultraPEALut administration. No adverse reaction was observed. Rat body weight was monitored during the experiment; thus, we have included in the text the result of the analysis of the cumulative body weight change following 14 consecutive co-ultra PEALut administrations compared to vehicle. We have also modified each of the result sections to highlight that no effect of PEALut alone was observed.
L57: The section focused on AD physiopathology mechanism hypothesis should be separate from “Aβ accumulation…”
We agree with the Reviewer, thus we followed her/his suggestion making the change.
L83: what are the medical purposes for glialia?
Glialia is a dietary food for special medical purposes, intended for the dietary management of individuals with central neuroinflammation and associated oxidative stress. We have added this information in the introduction.
L95 Author should add “i.p. administration”
Done
L112 role of astrocyte may include BBB structuration
We agree with the Reviewer and included her/his suggestion in the text.
L551, L574: DOI is missing. Please check all the bibliographic section
Following the Reviewer’s suggestion, we have checked the list of references and added several missing doi numbers.
Figures:
The legend of “veh/Aβ” using white or black rectangles is present just in Fig 1C and in Fig 3C, is missing in Fig 2. It should be easier to read aided with clear legend common for all the histograms.
According to the Reviewer’s suggestion, we have placed the legend in a more visible position in each figure.
Fig 1E: Y-axe should be normalized at 4 ΔΔCq
We have changed Fig.1E Y-axis as suggested.
Reviewer 2 Report
The topic of this paper is really interesting evaluating the possibility to early intervene on neurodegenerative disease with products active and safe. The present research is well planned and performed. Some minor points should be addressed:
- the authors are encouraged to improve the results section of abstract to offer a more clear picture of the study
- microglia and astrocytes were studied as Iba and GFAP expression but what about morphological features of these cells. Did the authors observe changes with Abeta or Pea?
- please specify in the results section that gene analysis was performed on hippocampus
- the authors state that Pealut was approved for human use as dietary food for special medical purposes, this justify the safety of the product but some comment about the toxicological profile could improve the manuscript and favor the clinical application
Author Response
We thank the Reviewer for the comments and appreciate her/his contribution to the improvement of the manuscript. Considering her/his criticisms, we have modified the manuscript. Please see the changes made in the main text as tracked change. We believe our manuscript has been greatly improved and we acknowledge the contribution of Reviewer#2.
the authors are encouraged to improve the results section of abstract to offer a more clear picture of the study
We agree with the Reviewer and, according her/his suggestion, we have added more details in the results section of the abstract.
microglia and astrocytes were studied as Iba and GFAP expression but what about morphological features of these cells. Did the authors observe changes with Abeta or Pea?
We thank the Reviewer for the possibility to discuss this key aspect. We observed important signs of activation of both astrocytes and microglia through the markers GFAP and Iba1, respectively. These two markers are often used to detect morphological changes. We observed an increased intensity of both signals in Abeta-injected rats (which usually correlates to altered morphology) and the higher resolution pictures in Fig. 1 show changes in the morphology of these cells. However, we could not perform an in-depth stereological analysis because we do not have the required software. For this reason, we preferred not to refer to morphological changes throughout the paper.
please specify in the results section that gene analysis was performed on hippocampus
According to the Reviewer’s suggestion, we have modified each section of the results in order to make clear that the analyses were performed in the hippocampus.
The authors state that Pealut was approved for human use as dietary food for special medical purposes, this justify the safety of the product but some comment about the toxicological profile could improve the manuscript and favor the clinical application
We thank the Reviewer for this important comment. Following her/his suggestion, we have added a sentence in the introduction and modified each of the result sections to highlight that no effect of PEALut alone was observed.